# Spectrum of sexual partner types among adults screened for sexually transmitted infections in the Eastern Cape, South Africa

Lindsey de Vos[1], Mandisa M.M. Mdingi[1,2]*, Ranjana M.S. Gigi[1,3], Avuyonke Gebengu[1], Remco P.H. Peters[1,2]

1 Research Unit, Foundation for Professional Development, East London, South Africa, 2 Department of Medical Microbiology, University of Pretoria, Pretoria, South Africa, 3 Institute of Social and Preventive Medicine, University of Bern, Bern, Switzerland

◉ These authors contributed equally to this work.
* MandisaM@foundation.co.za

## Abstract

In South Africa, *Chlamydia trachomatis* prevalence is 14.7% in women and 6.6% in men, while *Neisseria gonorrhoeae* rates are 6.0% and 3.4%. Partner management, including identifying and screening for STIs, is essential for STI control efforts but challenging due to relationship dynamics, fear of disclosure and stigma. This study aims to understand how adults in the Eastern Cape report partner types when seeking STI care, enhancing partner notification strategies and reducing transmission. From February-August 2023, a cross-sectional evaluation of a *Neisseria gonorrhoeae* lateral flow assay was conducted among asymptomatic adults aged 18–49 years at four primary healthcare facilities in Buffalo City, Eastern Cape. Participants completed surveys classifying partners into LUSTRUM team's 8 partner types and 5 type-classifications. Data were analyzed using StataSE 17, examining associations between partner types and variables like gender, location, number of sexual partners, and STI test results. A total of 500 men and 400 women (median ages 31 and 32) were recruited. The most reported partner types were main/serious/long-term partners (41%) and girlfriend/boyfriend (29%) or LUSTRUM's 4: established (67%) and occasional partners (28%). Participants reporting main/long-term partners, steady, or boyfriend/girlfriend showed variability in partner numbers. Male adults more commonly reported casual partner types like friends with benefits (21% vs. 9%) and fuck buddy/booty call (9% vs. 3%), with significant associations for super casual/hook-up/meet/one-night stand (p=0.02). Regional differences in partner types and a significant association between new partners and NG Xpert positivity (p=0.01) were observed. This study confirms the diverse spectrum of sexual partner types. Findings reveal that men and women may have different relationships, and regional variations suggest context-specific approaches are needed. Identifying partner types can enhance communication and treatment strategies and address significant gaps in partner notification and STI care.

**Data availability statement:** The data underlying the results presented in the study are available from the following OSF identifier DOI 10.17605/OSF.IO/9NPQK.

**Funding:** The NG-LFA performance evaluation ('Live Well' project) was supported by a FIND sub-award from the Global Antimicrobial Resistance Innovation Fund (GAMRIF).

**Competing interests:** The authors have declared that no competing interests exist.

## Introduction

WHO estimates that 374 million new curable sexually transmitted infections (STIs) are acquired worldwide every year [1]. The estimated prevalence for *Chlamydia trachomatis* (*C. trachomatis*) and *Neisseria gonorrhoeae* (*N. gonorrhoeae*) among women and men in South Africa was 14.7% and 6.0%, 6.6% and 3.4% respectively. The estimated annual incident cases for South Africa included 3.87 million for *C. trachomatis* and 2.21 million for *N. gonorrhoeae* [2]. The Eastern Cape region bears a significant burden of STIs nationally, with high incidences of male urethritis (MUS, 2,803 cases per 100 000) and vaginal discharge syndrome (VDS, 1,893 per 100 000) [3]. Furthermore, prevalence of *C. trachomatis* was 16% and *N. gonorrhoeae* 5% among a cohort of pregnant women attending antenatal care at the same healthcare facilities [4]. Untreated STIs can lead to genital tract morbidity, significant social and psychological consequences including stigma, and it is associated with reproductive tract complications, adverse birth outcomes, and increased risk of HIV transmission [1,5].

Improving access to targeted STI screening and treatment in low- to middle-income countries is essential for effective disease control [5]. Equally critical is the inclusion of sexual partners in care to curb risks of further onward transmission or re-infection. Partner management involves identifying, screening, and treating partners of STI-positive patients using various notification methods, such as patient or provider referral, enhanced notification with educational materials, and expedited partner treatment (partner pill packets) [6–10]. The optimal approach for partner notification in South Africa is unclear, and current methods like patient-delivered referral slips may be inadequate, particularly for casual partners who may react unpredictably to STI disclosure reducing effective notification and treatment [9,11–15].

Significant challenges in partner notification include discrepancies between the intention to disclose and actual STI disclosure due to factors like partner type, relationship quality, power dynamics, and communication issues [7,12–14]. Stigma, relational conflicts such as infidelity assumptions and shame, and worry for dissolution further deter partner notification, and may be compounded by a lack of STI counseling and relationship separations due to socio-economic reasons [15–17]. Women in South Africa face additional barriers such as partner substance use and fear of intimate partner violence, complicating STI disclosure [12,14,18,19].

Understanding sexual partner types may improve partner counseling and tailor relevant notification strategies by partner type [11,17]. Partner-type profiles, developed by Estcourt et al. (2022) and the LUSTRUM team, extend beyond 'steady' or 'casual' labels. The LUSTRUM's 8 partner type classification provides information about relationship duration, the likelihood of transmission/re-infection risks (including perceived risk and condom use), repeated sexual encounters, sexual mixing in networks/exclusivity, emotional connection, and contactability [20–22]. Based on expert input the classification was further refined to provide a better understanding of partner types for healthcare providers that may facilitate STI clinical management to include unique socio-psychological and behavioral factors [6,20,23].

To enhance person-centered STI partner notification, it is crucial to develop partner profiles that inform relationship dynamics and STI risk to guide partner management practice and to develop novel interventions for different partner types [20,21,23–25]. However, partner types may vary contextually, even within communities, and the socio-relational dynamics of partners may not be taken into consideration by healthcare workers to inform appropriate STI notification practices. This study investigates how adults in the Eastern Cape report partner types when seeking STI care at public healthcare facilities, aiming to better understand the spectrum of sexual partners in our setting with the ultimate goal to improve partner management strategies.

## Methods

### Study setting and sample

Between 22 February 2023 and 8 August 2023, a cross-sectional performance evaluation ('Live Well' project) was conducted for a novel *Neisseria gonorrhoeae* lateral flow assay (NG-LFA) among asymptomatic male and female adults aged 18–49 years. Previous studies have thoroughly detailed the NG-LFA administration and performance outcomes [26,27]. To assess the performance of the NG-LFA, in-facility STI testing for *C. trachomatis* and *N. gonorrhoeae* was performed as a comparison using a combined Xpert® CT/NG assay (Cepheid, Sunnyvale, CA) of male urine and healthcare-worker collected vaginal swab. In case of positive Xpert test result, treatment and partner notification slip was provided as per national STI management guidelines [28].

In brief, using convenience sampling, STI screening was offered to adult individuals presenting at four primary healthcare facilities in the Buffalo City Metropolitan (BCM) area, located in the Eastern Cape, South Africa. Presenting at the facility could include seeking healthcare services, accompanying others, or participating in the study after information was received in their community.

### Survey administration

Senior field workers recruited participants using stratified sampling (500 male and 400 female adults, aged 18–49 years, who seek any type of health care service, and are asymptomatic) and, following written informed consent, administered demographic and behavioural surveys at four healthcare sites (Site 1–4). The study did not have any specific exclusion criteria, except for age (individuals younger than 18 years). The survey covered socio-demographic data, sexual partnerships, and clinical presentation including STI history and HIV status. Socio-demographic data collected via REDCap by research assistants include the following standard variables for this research setting: age, recruitment facility, gender identity (regardless of sex at birth), education level (none to tertiary), income source (formal employment, self-employment, student, unemployed), and the number of sexual partners in the past 6 months. Additionally, research nurses gathered clinical data on symptoms (STIs or others), prior treatment for VDS/MUS, reported condom use and HIV status.

The survey also employed partner classification systems as developed by Estcourt et al. (2022) [20]. The two included classification systems included LUSTRUM's 8 partner types: 1) married/committed, 2) main partner/serious/stable/long-term partner, 3) steady, 4) girlfriend/boyfriends, 5) dating/going out, 6) friends with benefits, 7) fuck buddies/booty calls, and 8) super casual/hook-up/meet/one night stand and LUSTRUM's 5 refined partner types originally developed for clinical application to identify and categorize partners: 1) established partner, 2) new partner, 3) occasional partner, 4) one-off partner, 5) sex worker [20]. Participants were able to select multiple partner types depending on the number of partners reported. Each partner type reveals key biophysical, psycho-social features that are different including perceived exclusivity, sexual frequency and reinfection risks, duration or countability. For instance, sexual exclusivity or the emotional connection may be slightly higher among partners defined as 'steady' versus a boyfriend/girlfriend. Detailed explanations that differentiate these partner types has been described by the LUSTRUM team elsewhere [20]. As sex workers was not selected by this population, this was dropped from the analysis and this classification was herein referred as 'LUSTRUM's 4'.

## Data analysis

Data was captured using REDCap (Research Electronic Data Capture) [29,30]. Participant's sociodemographic and partner characteristics for this sub-study were analysed using StataSE 17 (StataCorp, LLC, College Station, TX) and presented as median or counts/proportions. Further associations between partner types and other variables such as gender, location, the number of sexual partners, and STI test results were also analyzed using logistic regression. Gender and number of partners were statistically analyzed as they may inform differences in partner types and STI outcomes. Married/committed and established partners were included as reference level for each classification. Analysis was controlled for number of partners (one or more partners). Since all participants provided responses to both the 8- and 5-type partner classification systems, a comparison was conducted to assess the degree of overlap between the two systems as reported by participants.

## Ethics

Approval was obtained from the Faculty of Health Sciences Research Ethics Committee at the University of Pretoria (510/2021). Participants were informed and provided written consent to the study using an informed consent form. Participants were reimbursed with a R100 (~$5.4) grocery voucher for their participation.

## Results

### Participant characteristics

A total of 500 men and 400 women were recruited, median age was 31 (IQR 23–38) and 32 (IQR 24–39) respectively (Table 1). Participants were primarily those visiting the healthcare facility as bystanders or being referred from the community (Males; 78% and Females; 55%).

Men were more likely to report having multiple partners (more than one) in the past 6 months (57%) compared to women (26%). Reported condom use at the last sexual act was slightly higher among male (37%) versus female (31%) participants.

### Partner type categorization

**LUSTRUM's 8 partner types.** The most reported partner types were main/serious/stable/long-term partners (n = 373, 41%), and girlfriend/boyfriend (n = 261, 29%). Dating/going out was not commonly used amongst this population (n = 14, 1.6%).

Reported partner types were strongly correlated with the number of sexual partners. As shown in Table 2, there were stronger associations between the number of sexual partners and casual definitions (p < 0.001), versus steady or dating relationships.

Fig 1 and Table 3 show partner type by gender, adjusted for number of partners. More casual partner typing was found amongst male adults, including friends with benefits (21% vs 9%), and fuck buddy/booty call (9% vs 3%). A significant association was found between gender and super casual/hook-up/one-night stand (p = 0.02).

**LUSTRUM's 4 partner types.** Most frequently reported types were established (n = 599, 67%) or occasional partners (n = 255, 28%). Established partners were more common among those with one sexual partner (74%), whereas occasional partners were more frequently reported among those with multiple partners (59%) (Table 4). All partner types were significantly associated with the number of sexual partners (one or more partners).

Similarly, when assessing LUSTRUM's 4 partner type classification by gender (Table 5), controlled for number of partners, more male participants defined partners as occasional (38% versus 16%) or once-off partners (18% versus 4%) and this association was also significant (p < 0.001).

**Table 1. Characteristics of male and female participants of sexually transmitted infections screening study.**

| Variable | | Male (n = 500) | Female (n = 400) |
|---|---|---|---|
| **Age** (median, IQR) | | 31 (23-38) | 32 (24-39) |
| **Reported reasons for presenting at the health-care facility** | Antenatal care | 0 (0) | 4 (1.0) |
| | Antiretroviral therapy | 39 (7.8) | 93 (23.3) |
| | SRH | 12 (2.4) | 26 (6.5) |
| | Other Chronic | 14 (2.8) | 19 (4.8) |
| | Acute | 40 (8.0) | 36 (9.0) |
| | PrEP | 3 (0.6) | 4 (1.0) |
| | Bystander/Community | 392 (78.4) | 218 (54.5) |
| **Highest education level completed**[a] | None | 7 (1.4) | 4 (1.0) |
| | Primary | 234 (47.9) | 169 (42.7) |
| | Matric | 210 (42.9) | 178 (45.0) |
| | Tertiary | 38 (7.8) | 45 (11.4) |
| **Source of income**[b] | Formal employment | 86 (18.0) | 89 (23.5) |
| | Self-employed | 31 (6.5) | 18 (4.8) |
| | Student | 44 (9.2) | 43 (11.4) |
| | Unemployed | 316 (66.3) | 228 (60.3) |
| **Sexual partners past 6 months** | 0 | 10 (2.0) | 18 (4.5) |
| | 1 | 198 (40.0) | 276 (69.5) |
| | 2 | 115 (23.2) | 69 (17.4) |
| | 3-5 | 135 (27.3) | 31 (7.8) |
| | 6 or more | 37 (7.5) | 3 (0.8) |
| **Reported condom use last sexual contact**[c] | Yes | 181 (36.6) | 81 (20.5) |
| | No | 313 (63.4) | 314 (79.5) |
| **Ever treated for STIs (MUS/VDS)** | No | 325 (69.0) | 231 (61.8) |
| | Yes, < 1 month ago | 11 (2.3) | 15 (4.0) |
| | Yes, < 12 months ago | 44 (9.3) | 43 (11.5) |
| | Yes, longer ago | 91 (19.3) | 85 (22.7) |
| **Self-reported HIV status** | Positive | 58 (11.6) | 111 (27.8) |
| | Negative | 413 (82.6) | 282 (70.5) |
| | Unknown | 26 (5.2) | 6 (1.5) |
| | Did not disclose | 3 (0.6) | 1 (0.3) |

SRH, Sexual and reproductive health; MUS, Male urethritis syndrome; VDS, Vaginal discharge syndrome.

[a]N = 489 males and N = 396 females reported education level.

[b]N=477 males and N=378 females reported on sources of income.

[c]N = 494 males and N = 395 females responded condom use at last sexual contact.

Fig 2 compares partner types across LUSTRUM's 8 versus 4 partner type classifications. Both classifications strongly correlate, participants who indicated having married/committed, serious, or steady partners also selected 'established partner'. A similar trend was observed for casual partner definitions in both classifications. For instance, 'friends with benefits' and 'fuck buddy/booty call' aligned with 'occasional partner' types, while 'super casual/hook up/ meet/one night stand' corresponded most often with 'once-off' partners. More variability can be noted for partners considered main/serious or long-term, steady or boyfriend/girlfriend, where they are also defined as established, new, occasional, or once-off.

**Table 2. Sexual types by participants reporting one or more partners.**

| Type | One partner (n = 474) | Two or more partners (n = 390) | P |
|---|---|---|---|
| Married/ Committed | 49 (10.3) | 14 (3.6) | |
| Main partner/ serious/ stable/ long-term partner | 208 (43.9) | 164 (42.1) | 0.17 |
| Steady | 110 (23.2) | 90 (23.1) | 0.53 |
| Girlfriend/ Boyfriend | 137 (28.9) | 121 (31.0) | 0.61 |
| Dating/ Going out | 3 (0.6) | 10 (2.6) | **0.04** |
| Friends with benefits | 14 (3.0) | 128 (32.8) | **<0.001** |
| Fuck buddy/ Booty call | 4 (0.8) | 53 (13.6) | **<0.001** |
| Super casual/ hook-up/ meet/ one-night stand | 11 (2.3) | 160 (41.0) | **<0.001** |

*28 participants reported no sexual partner in the past 6 months, n = 8 missing value for sexual partner.

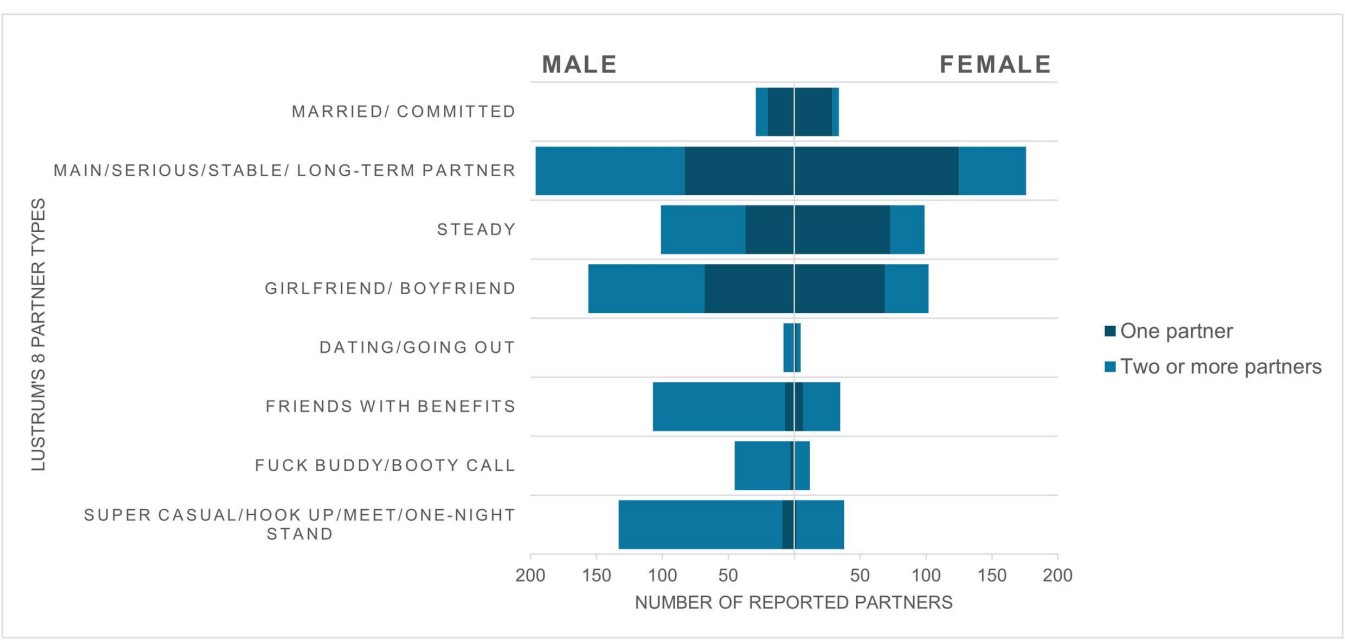

**Fig 1. Partner types reported by gender, adjusted for number of partners.** Dark blue - one partner; blue - two or more partners.

**Table 3. LUSTRUM's 8 partner types by gender.**

| Type | Female (n = 400) | Male (n = 500) | P |
|---|---|---|---|
| Married/ Committed | 34 (8.5) | 29 (5.8) | – |
| Main partner/ serious/ stable/ long-term partner | 177 (44.3) | 196 (39.2) | 0.07 |
| Steady | 100 (25.0) | 103 (20.6) | **0.04** |
| Girlfriend/ Boyfriend | 104 (26.0) | 157 (31.4) | 0.13 |
| Dating/ Going out | 6 (1.5) | 8 (1.6) | 0.70 |
| Friends with benefits | 35 (8.8) | 107 (21.4) | 0.13 |
| Fuck buddy/ Booty call | 12 (3.0) | 45 (9.0) | 0.17 |
| Super casual/ hook-up/ meet/ one-night stand | 39 (9.8) | 133 (26.6) | **0.02** |

**Table 4. Sexual partner types reported by those who have one or more partners.**

| Type | One partner (n = 474) | Two or more partners (n = 390) | P |
|---|---|---|---|
| Established partner | 350 (73.8) | 247 (63.3) | – |
| New partner | 15 (3.2) | 50 (12.8) | **<0.001** |
| Occasional partner | 24 (5.1) | 231 (59.2) | **<0.001** |
| Once-off partner | 10 (2.1) | 97 (24.9) | **<0.001** |

**Table 5. LUSTRUM's 4 - Partner types as reported by female and male adults (past 6 months) controlled for number of partners (one or more).**

| Type | Female (n = 400) | Male (n = 500) | p |
|---|---|---|---|
| Established partner | 286 (71.5) | 313 (62.6) | – |
| New partner | 24 (6.0) | 43 (8.6) | 0.33 |
| Occasional partner | 63 (15.8) | 192 (38.4) | 0.06 |
| Once-off partner | 17 (4.3) | 92 (18.4) | **<0.001** |

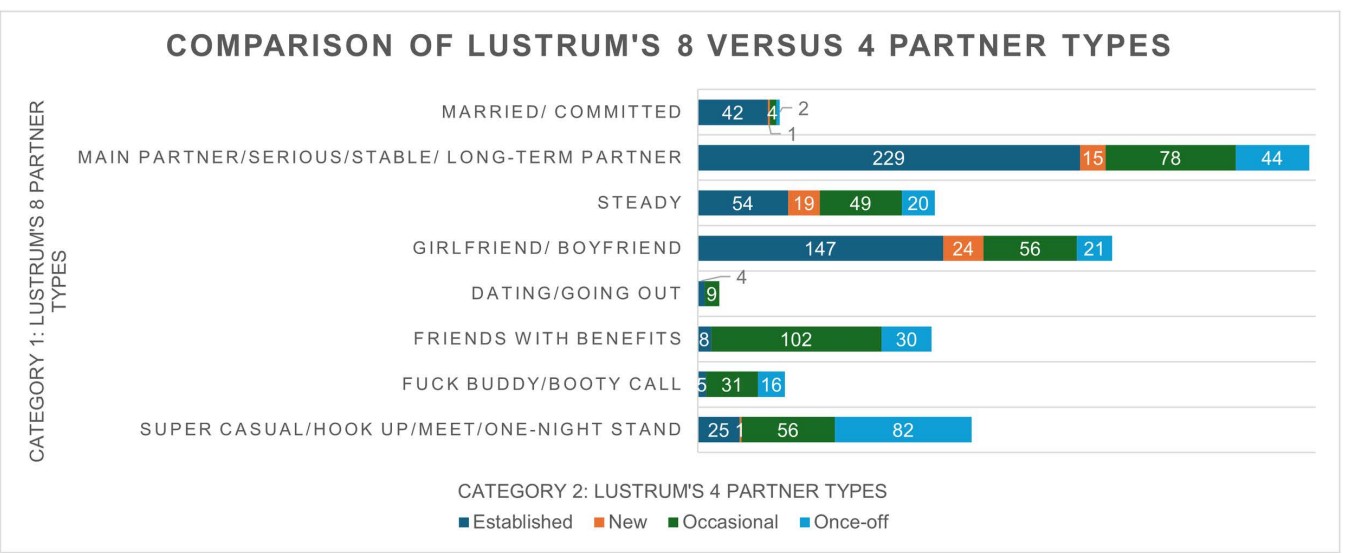

**Fig 2. A comparison between participants reporting LUSTRUM's 8 and LUSTRUM's 4 partner types.**

## LUSTRUM's 8 sexual partners by geographic location

In the rural site (Site 1), 'girlfriend/boyfriend' was the predominant partner type (n = 178/262, 68%) compared to the other sites (Table 6). Additionally, super casual/hook-up/meet or one-night stands were more common compared to other regions (n = 75/262, 29%). At Site 2, which also largely serves an informal settlement, main/serious/stable/long-term partners (n = 176/321) and friends with benefits (n = 87/321) were more frequently reported. In contrast, at the peri-urban sites, partners were more often identified as 'steady' (n = 133, 42%).

**Table 6. Sexual partner types by study region.**

|  | Site 1, rural (n = 262) | Site 2, township (n = 321) | Site 3 & 4, peri-urban (n = 317) |
|---|---|---|---|
| Married/ Committed | 25 (9.5) | 16 (5.0) | 22 (6.9) |
| Main partner/ serious/ stable/ long-term partner | 96 (36.6) | 176 (54.8) | 101 (31.9) |
| Steady | 25 (9.5) | 45 (14.0) | 133 (42.0) |
| Girlfriend/Boyfriend | 178 (67.9) | 32 (10.0) | 51 (16.0) |
| Dating/ Going out | 6 (2.3) | 4 (1.3) | 4 (1.3) |
| Friends with benefits | 22 (8.4) | 87 (27.1) | 33 (10.4) |
| Fuck buddy/ Booty call | 19 (7.3) | 19 (5.9) | 19 (6.0) |
| Super casual/ hook-up/ meet/ one-night stand | 75 (28.6) | 61 (19.0) | 36 (11.4) |

## Partner type and STI results

To determine whether there are differences between partner type and individuals presenting with and without an STI we compared definitions per LUSTRUM's 4 partner type classification and participant's Xpert CT/NG test results (S1 Table). *C. trachomatis* and *N. gonorrhoeae* positivity was highest amongst men reporting new partners (14% and 18.6%). Among female participants, the *C. trachomatis* prevalence was higher among those reporting once-off partners (18%) and *N. gonorrhoeae* was higher among those reporting new partners (13%) although samples were small. *N. gonorrhoeae* positivity was significantly associated with new partners.

## Discussion

This study is one of the few examining the types of sexual partners among male and female adults accessing primary healthcare services and screening for STIs. Originally developed in the UK, the sexual partner classifications have been applied in South Africa, providing valuable insights for public health and partner notification strategies. Our findings reveal a strong correlation between the two classifications, supporting the refinement from 8 (married/committed, main partner/serious/stable/long-term partner, 3) steady, girlfriend/boyfriends, dating/going out, friends with benefits, fuck buddies/booty calls, and super casual/hook-up/meet/one night stand) to 5 partner types (established partner, new partner, occasional partner, one-off partner, where sex workers was omitted due to none reported) by the LUSTRUM team for healthcare use [6,20]. Similar to the LUSTRUM trial, the most common partner types in this study were distinguished as established and occasional. However, participants highlighted through LUSTRUM 8 that there is variability in how these partner types are defined, particularly when distinguishing between main partners, steady partners, and girlfriend/boyfriend (Fig 2). More cases of friends with benefits, fuck buddies and super casuals or hookups are reported as occasional partners. Similarly, the assumption is that main/long-term partners might equate to established or one partner only. In contrast, our findings show that these participants can have multiple partners that healthcare providers could potentially miss.

Males and females show different relationships, with males more often reporting them as casual. The variation in partner types reported between rural, peri-urban, and township settings in our study suggests different approaches based on urban or rural settings. Participants identified friends with benefits or fuck buddies more often as occasional partners, and hookups as one-time partners. Those in multiple partnerships reported both established and casual partners, highlighting the need to include all partner types in STI screening. Positive Xpert test results among those with new partners in the past six months may help understand transmission risks by partner type, though this trend was not seen for *C. trachomatis* [25,31].

The high number of adults reporting multiple and casual partners underscores the possible challenges of the current partner notification system. Disclosure of STIs is shown to correlate with the number of sexual partners, with individuals

with fewer partners being more inclined to disclose [32]. Further, although most pregnant women disclose STIs to their partners, less than half of the partners receive treatment. These findings also highlight the barriers associated with partner involvement if the dynamics of the relationship are not fully understood, as failure to adequately notify partners undermines the effectiveness of STI testing during this vulnerable period [9,33,34]. In the LUSTRUM's trial, established or new partners were more likely to be treated for *C. trachomatis* within 2–4 weeks compared to once-off partners [6]. Identifying partner types at STI screening may inform how STIs are communicated within relationship types or whether partners are likely to actively seek treatment or not (depending on the strategy used) [11,14,16]. Male adults report more casual partner types which may explain one of the contributing reasons why fewer men disclose to sexual partners as compared to women [12,16]. Lack of partner treatment was associated with repeat infections [10,33], underscoring adequate notification and referral as the largest gap in adequate STI care [16].

## Implications

At the same time, the feasibility of partner management differs between partner types. Findings show that given differences in partner profiles programs should be context-specific for impact. Although partner treatment is crucial, enhanced or expedited partner referral or alternative strategies are not currently supported in South Africa [9]. These strategies may help increase the chances of partners being screened, reduce the time it takes to access care, and ease the pressure on the index patient when it comes to partner testing. Strategies could include options like home screening kits, additional STI counseling and health education for both partners to raise awareness and reduce reinfection rates, beyond partner notification slips alone. Other options might involve healthcare professionals directly notifying partners or using electronic messaging to address disclosure concerns. The findings from these strategies could also help determine which strategies are most suitable based on factors like partner type, the gender of the index patient, and the number of partners involved [6,8,10,18,19].

## Limitations

The existing partner types may overlook important cultural aspects and definitions specific to this study area that are not captured by the current framework. However, the partner type categorizations developed and defined by LUSTRUM provided sufficient variability and representation of different partner types. Each classification system was aligned, particularly in aligning the 8-partner types as more established and/or occasional partners according to LUSTRUM-4. The number of partners was statistically significant with the 4-partner type categorization. However, this study did not allow participants to report partner types outside the predefined categories or as "other." Varying methods such as qualitative interviews or focus groups could be used to explore how men and women in this context describe current sexual partnerships and how these are influenced by existing social and cultural factors [23]. Although participants were recruited from a specific health district, they were a representative sample of the general population who were asymptomatic for STIs. The study prevalence reflected the estimated national population prevalence of 5% (3% among men; 7% in women). Therefore, a sample size of 900 participants was sufficient to determine the diagnostic performance and the target product profile requirements of the NG-LFA. Although associations were found between male gender reporting new partners and *N. gonorrhoeae*, positive test result numbers were small. The categorization of partner types were administered by senior field workers to clarify partner definitions before STI screening. However, having professional nurses who counsel on STIs and partner notification, and provide treatment at the primary healthcare level, classify partner relationships at the time of diagnosis (as done in the LUSTRUM trial) could improve future partner notification strategies, going beyond just partner classifications. A deeper understanding of partner relationships and dynamics could be achieved by comparing the definitions provided by index patients with those of their partners.

## Conclusions

Findings confirm that partner definitions among adults seeking STI-related care extend beyond a simple dichotomy. These definitions are further influenced by factors such as the number of sexual partners and geographic location. A deeper understanding of cultural definitions and the differences in partner interpretations by factors such as gender could also help understand current gaps or difficulties with partner notification. Identifying partner types at screening encourages relationship-dynamic discussion to further inform effective STI prevention and care.

## Supporting information

**S1 Table. LUSTRUM's 4 partner types and Xpert CT/NG positivity.**
(PDF)

## Acknowledgments

Study permission was given by the Eastern Cape Department of Health and Buffalo City Metropolitan Health District. We would like to thank the FPD STI research staff and participants for their contributions to this study.

## Author contributions

**Conceptualization:** Remco P.H. Peters.

**Formal analysis:** Lindsey de Vos.

**Funding acquisition:** Remco P.H. Peters.

**Investigation:** Mandisa M.M. Mdingi, Ranjana M.S. Gigi, Remco P.H. Peters.

**Project administration:** Mandisa M.M. Mdingi, Ranjana M.S. Gigi.

**Supervision:** Remco P.H. Peters.

**Validation:** Remco P.H. Peters.

**Visualization:** Lindsey de Vos.

**Writing – original draft:** Lindsey de Vos.

**Writing – review & editing:** Mandisa M.M. Mdingi, Ranjana M.S. Gigi, Avuyonke Gebengu, Remco P.H. Peters.

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
