## [Decision Letter · Decision Letter 0]

20 Feb 2025

PONE-D-24-26246Spectrum of sexual partner types among adults screened for sexually transmitted infections in the Eastern Cape, South AfricaPLOS ONE

Dear Dr. Mdingi,

Thank you for submitting your manuscript to PLOS ONE. After careful consideration, we feel that it has merit but does not fully meet PLOS ONE’s publication criteria as it currently stands. Therefore, we invite you to submit a revised version of the manuscript that addresses the points raised during the review process.

We look forward to receiving your revised manuscript.

Kind regards,

Caroline Watts, PhD

Academic Editor

PLOS ONE

Journal Requirements:

2. We note that your Data Availability Statement is currently as follows:

“All relevant data are within the manuscript and its Supporting Information files.”

Please confirm at this time whether or not your submission contains all raw data required to replicate the results of your study. Authors must share the “minimal data set” for their submission. PLOS defines the minimal data set to consist of the data required to replicate all study findings reported in the article, as well as related metadata and methods (https://journals.plos.org/plosone/s/data-availability#loc-minimal-data-set-definition ).

If your submission does not contain these data, please either upload them as Supporting Information files or deposit them to a stable, public repository and provide us with the relevant URLs, DOIs, or accession numbers. For a list of recommended repositories, please see https://journals.plos.org/plosone/s/recommended-repositories .

Additional Editor Comments:

Please note reviewer comments regarding tables and look at merging tables where there are similar variables ie Table 1 (second), 2 and 5 and Table 3 and 4.

Reviewers' comments:

Reviewer's Responses to Questions

**Comments to the Author**

1. Is the manuscript technically sound, and do the data support the conclusions?

Reviewer #1: Yes

Reviewer #2: Yes

2. Has the statistical analysis been performed appropriately and rigorously?

Reviewer #1: Yes

Reviewer #2: I Don't Know

3. Have the authors made all data underlying the findings in their manuscript fully available?

Reviewer #1: Yes

Reviewer #2: Yes

4. Is the manuscript presented in an intelligible fashion and written in standard English?

Reviewer #1: Yes

Reviewer #2: Yes

5. Review Comments to the Author

Reviewer #1: Thank you for providing the opportunity to review this interesting manuscript. Improved partner notification is increasingly important to control STI transmission. Defining relationship types may be crucial in identifying risks and developing optimal partner notification strategies. By using two kinds of classification, you examined the type of partner relationships that were most common among STI clinic-visiting participants in South Africa. In addition, you suggested improvements for the LUSTRUM classification based on cultural aspects.

I do have some suggestions and remarks for the author to consider, you can find them below:

Introduction

• In line 36 you mentioned that the majority of STIs occur asymptomatic, followed by lines 37-39 where you described the complications of untreated STIs. A quick reader may link asymptomatic STI directly to complications, however, recent studies are questioning whether to continue testing and treating asymptomatic C.Trachomatis (https://www.ncbi.nlm.nih.gov/pmc/articles/PMC8922931/). Could you adjust these sentences, while being cautious of this current knowledge?

Methods

• Please write the dates out in words (e.g., January 1, 2001)

• Lines 87 – 91 regarding the epidemiology of C. Trachomatis and N. Gonorrhoea may be replaced (preferably to the introduction) to enrich background information on STI burden in the Eastern Cape.

• In line 93 you briefly described the recruitment process. Could you add in- and exclusion criteria?

• The method section seemed incomplete. I have not discovered a paragraph with measurements and definitions of your variables such as the socio-demographic data, which questions did you ask the patients to get this information? For instance, is gender defined as a male assigned at birth or identification of the patient with male gender? What were the answering options for participants? The same goes for the STI clinic sites (1-4, what does it entail?) and level of education; please specify this in the method section.

• Lines 105-108: Could you elaborate on why you have chosen to only consider genital infections of C.Trachomatis and N.Gonorrhoea?

Results

• Line 126: Please add (if available) information on the sexual orientation (e.g., MSM, heterosexual) of the participants.

• Please validate why you included individuals who did not have any sexual partners in the past 6 months.

• Consider adding (statistical) validation as to why you report men and women separately in various tables. When you do, please add it to the method section.

• Tables are not numbered correctly; there are 2 tables named ‘table 1’.

• First table has “Male” placed in the right column, however in the other tables “Male” is placed in the right column. Please transform to stay consistent.

• Consider highlighting significant p-values, for instance in a bold font (Note: only when in alignment with the journals’ guidelines)

• Line 158: In the method section you proposed the use of the “LUSTRUM 5” types. However, this paragraph’s title immediately raises questions. Even though an explanation is given in lines 159 and 160, this explanation may be more fitting in the method section.

• Could you explain why you only assessed STI association (supplementary table) with the LUSTRUM 4 and not with the LUSTRUM 8 classification?

Discussion

• Please reflect on findings from Figure 2, since you are raising important insights in the use of these different LUSTRUM classifications.

• Line 219: Write out “wasn’t”

• In line 224 a link with pregnant women is made, regarding to disclosure of STI status to their partner. Could you clarify how the referenced studies are linked to the results of this study?

• Future strategies described in lines 239 – 242 may benefit from further explanation. For instance, clarify how e-Health may enhance partner notification or accessibility into STI care.

• Lines 244-245 mentioned a limitation of the current LUSTRUM framework when applied in South African culture. Please consider adding a separate ‘Implications’ paragraph where you reflect on this conclusion and state potential adjustments for future research.

• Lines 248-249: Could you indicate (statistical) evidence of representativeness of your study sample?

• A limitation in line 252 is the need for professional nurses when defining partner relationships. Could you explain if this is feasible in real-world applications of partner notification strategies?

Reviewer #2: Review of “Spectrum of sexual partner types among adults screened for sexually transmitted

infections in the Eastern Cape, South Africa”

Summary of review

The authors nicely outline their experience with assessing partner types as a part of a larger STI study. It would be helpful to better define the LUSTRUM partner definitions and explain how the partner types differed from partner type categorization to better orient the reader. It would also increase the impact of your work if you could further tie in your thoughts regarding how well the LUSTRUM definitions worked in your contexts and what steps if any would help propel the field forward in your context.

Overall

1. I recommend maintaining one term - partner notification vs. disclosure - throughout the manuscript for consistency.

Abstract

1. Please specify the term “rates” in the first paragraph. From the Introduction this looks like the prevalence?

2.

Methods

1. Lines 87-91 are not Methods and better placed in the Introduction.

2. How do the partner types and partner categorizations differ?

3. A definition of each partner type and partner categorization would be helpful – for example, it is not clear to me how girl/boyfriend vs. steady differ.

4. Under data analysis, the authors write, “A comparison was conducted between the two partner classification systems.” How was this comparison made?

5. Who underwent what testing? It looks like participants underwent Xpert testing and also the parent study NG-LFA? It would be helpful to combine the testing sentences into one section.

Results

1. Line 126, when describing age, please label the age ranges with IQR.

2. Line 127-128 – discussion of the participant recruitment methods should fall under the Methods section. Please provide more detail on the recruitment. “recruited by accompanying patients to the facility” and “through referral from the community” are vague.

3.

Discussion

1. The authors write in lines 206-207, “Our findings reveal a strong correlation between the two classifications, supporting the refinement from 8 to 5 categories for healthcare use” What categories are these?

2. I’m interested to know how the authors consider the LUSTRUM definitions in the South African context. Did these definitions hold? Were there gaps in the definitions? What work (if any) is needed in this space in your local context?

Figures/Tables

1. I recommend defining the Table 1 terms for visiting clinic both in the table and in the Methods section.

2. Figure 1 is unnecessary since Table 2 encompasses this info with more detail.

6. PLOS authors have the option to publish the peer review history of their article (what does this mean? ). If published, this will include your full peer review and any attached files.

**Do you want your identity to be public for this peer review?** For information about this choice, including consent withdrawal, please see our Privacy Policy .

Reviewer #1: No

Reviewer #2: No

---

## [Author Response · Author response to Decision Letter 1]

26 Mar 2025

JOURNAL REQUIREMENTS

https://journals.plos.org/plosone/s/file?id=wjVg/

PLOSOne_formatting_sample_main_body.pdf and

https://journals.plos.org/plosone/s/file?id=ba62/

PLOSOne_formatting_sample_title_authors_affiliations.pdf We thank the editorial team and reviewers for their feedback and their time revising our manuscript.

We have ensured that the manuscript meets PLOS ONE's style requirements, including the title page and main text.

2. We note that your Data Availability Statement is currently as follows:

“All relevant data are within the manuscript and its Supporting Information files.”

If there are ethical or legal restrictions on sharing a de-identified data set, please explain them in detail (e.g., data contain potentially sensitive information, data are owned by a third-party organization, etc.) and who has imposed them (e.g., an ethics committee). Please also provide contact information for a data access committee, ethics committee, or other institutional body to which data requests may be sent. If data are owned by a third party, please indicate how others may request data access. We have taken note of the data availability requirements. We have made the minimal data set upon which this analysis/report is based available through the following DOI 10.17605/OSF.IO/9NPQK.

3. Please note reviewer comments regarding tables and look at merging tables where there are similar variables ie Table 1 (second), 2 and 5 and Table 3 and 4. Thank you, we have taken note of this suggestion, and have also noted that table numbers were mislabeled, and apologize for this oversight.

REVIEWER COMMENTS

Reviewer #1

4. Thank you for providing the opportunity to review this interesting manuscript. Improved partner notification is increasingly important to control STI transmission. Defining relationship types may be crucial in identifying risks and developing optimal partner notification strategies. By using two kinds of classification, you examined the type of partner relationships that were most common among STI clinic-visiting participants in South Africa. In addition, you suggested improvements for the LUSTRUM classification based on cultural aspects.

I do have some suggestions and remarks for the author to consider, you can find them below: We are pleased to hear that the reviewers found the manuscript interesting and that it raises important discussion points particularly relating to partner notification and STI control.

5. INTRODUCTION

In line 36 you mentioned that the majority of STIs occur asymptomatic, followed by lines 37-39 where you described the complications of untreated STIs. A quick reader may link asymptomatic STI directly to complications, however, recent studies are questioning whether to continue testing and treating asymptomatic C. Trachomatis (https://www.ncbi.nlm.nih.gov/pmc/articles/PMC8922931/). Could you adjust these sentences, while being cautious of this current knowledge? We appreciate the reviewers' feedback. For clarity, we have removed the sentence originally 37-39, now line 40-41 in reference to diagnostic tests. However, we still consider complications related to symptomatic STIs to be relevant.

6. METHODS

Please write the dates out in words (e.g., January 1, 2001) We have written out the duration of the study as 22 February 2023 and 8 August 2023 in line 83

7. Lines 87 – 91 regarding the epidemiology of C. Trachomatis and N. Gonorrhoea may be replaced (preferably to the introduction) to enrich background information on STI burden in the Eastern Cape. We have moved this information to the introduction in Line 36-40

8. In line 93 you briefly described the recruitment process. Could you add in- and exclusion criteria? We have clarified the inclusion and exclusion criteria as follows in lines 104-108: “Senior field workers recruited participants using stratified sampling (500 male and 400 female adults, aged 18 to 49 years, who seek any type of health care service, and are asymptomatic) and, following written informed consent, administered demographic and behavioural surveys at four healthcare sites (Site 1-4). The study did not have any specific exclusion criteria, except for age (individuals younger than 18 years).”

9. The method section seemed incomplete. I have not discovered a paragraph with measurements and definitions of your variables such as the socio-demographic data, which questions did you ask the patients to get this information? For instance, is gender defined as a male assigned at birth or identification of the patient with male gender? What were the answering options for participants? The same goes for the STI clinic sites (1-4, what does it entail?) and level of education; please specify this in the method section. We have included more clarity on the measurements and definitions of variables as follows;

Lines 109-115; “Socio-demographic data collected via REDCap by research assistants include the following standard variables for this research setting: age, recruitment facility, gender identity (regardless of sex at birth), education level (none to tertiary), income source (formal employment, self-employment, student, unemployed), and the number of sexual partners in the past 6 months. Additionally, research nurses gathered clinical data on symptoms (STIs or others), prior treatment for VDS/MUS, reported condom use and HIV status.”

10. Lines 105-108: Could you elaborate on why you have chosen to only consider genital infections of C.Trachomatis and N.Gonorrhoea? In the methods we describe that the partner data was part of a cross-sectional performance evaluation for a novel Neisseria gonorrhoeae lateral flow assay (NG-LFA). The GeneXpert CT/NG is a combined test, used as the gold-standard for the performance evaluation, thus, both these genital infections were considered for participants. We have added some details for clarity in line 87-91.

11. RESULTS

Line 126: Please add (if available) information on the sexual orientation (e.g., MSM, heterosexual) of the participants. Unfortunately we did not collected sexual orientation as part of this survey.

12. Please validate why you included individuals who did not have any sexual partners in the past 6 months. Participants were asked to self-report the number of sexual partners, which may have resulted in either over- or under-reporting. The reported number of partners could provide valuable insights into potential infection rates. For example, individuals who did not have a partner in the past six months may still test positive for infections.

13. Consider adding (statistical) validation as to why you report men and women separately in various tables. When you do, please add it to the method section. During our analysis we wanted to determine whether there was an interaction/difference between gender and the classification of partner types. This may also inform why there are STI differences by gender.

We have added a justification in the methods section. Lines 143-144; “Gender and number of partners were statistically analyzed as they may inform differences in partner types and STI outcomes.”

14. Tables are not numbered correctly; there are 2 tables named ‘table 1’. We apologize for the oversight; we have made the necessary revisions.

15. First table has “Male” placed in the right column, however in the other tables “Male” is placed in the right column. Please transform to stay consistent. We appreciate this suggestion. For consistency, we have placed ‘Males’ in the same column for each table.

16. Consider highlighting significant p-values, for instance in a bold font (Note: only when in alignment with the journals’ guidelines) Thank you, we have taken note and bolded the p-values.

17. Line 158: In the method section you proposed the use of the “LUSTRUM 5” types. However, this paragraph’s title immediately raises questions. Even though an explanation is given in lines 159 and 160, this explanation may be more fitting in the method section. Although this was only identified during the analysis process, we are happy to move the justification to the methods, to inform why one of the categories was dropped.

Lines 117-123

18. Could you explain why you only assessed STI association (supplementary table) with the LUSTRUM 4 and not with the LUSTRUM 8 classification? The 8 partner type was the initial classification system developed by the LUSTRUM team, that was comprehensive in capturing the biomedical and psychosocial differences between these partner types. However, these 8 categories were reduced to five, based on what was more practical for implementation and decision-making in clinical care, as well as expert input (Estcourt, 2021). Experts recommended including sex workers but this was not reported in our study sample, thus it was reduced and named LUSTRUM 4 for this study. As the revised 5 partner-type classification was to accommodate healthcare professionals and clinical care, we felt it was best suited to use this classification to assess STI associations, while still allowing for a more comprehensive classification with 8 categories for other associations, to assist with might be most relevant to our context, based on a comprehensive classification system.

19. DISCUSSION

Please reflect on findings from Figure 2, since you are raising important insights in the use of these different LUSTRUM classifications. Thank you for this comment. We have made these revisions to reflect on findings from Fig 2 and included these as follows in line 245-249: “Similar to the LUSTRUM trial, the most common partner types in this study were established and occasional. However, participants highlighted in LUSTRUM 8 that there is variability in how these partner types are defined, particularly when distinguishing between main partners, steady partners, and girlfriend/boyfriend (Fig 2). More cases of friends with benefits, fuck buddies and super casuals or hookups are reported as occasional partners”

20. Line 219: Write out “wasn’t” We have corrected this as ‘was not’ in line 262

21. In line 224 a link with pregnant women is made, regarding to disclosure of STI status to their partner. Could you clarify how the referenced studies are linked to the results of this study? Due to the nature of basic antenatal care at primary healthcare level, women are regularly screened and treated syndromically for STI symptoms including partner notification. We have further justified the link made to pregnant women and the inclusion of these references:

Line 268-271: “These findings also highlight the barriers associated with partner involvement if the dynamics of the relationship are not fully understood, as failure to adequately notify partners could increase the risk of STI reinfection and undermine the effectiveness of STI testing during this vulnerable period”

22. Future strategies described in lines 239 – 242 may benefit from further explanation. For instance, clarify how e-Health may enhance partner notification or accessibility into STI care. We have elaborated on the future strategies and potential impacts in lines 287-295:

“These strategies may help increase the chances of partners being screened, reduce the time it takes to access care, and ease the pressure on the index patient when it comes to partner testing. Strategies could include options like home screening kits, additional STI counseling and health education for both partners to raise awareness and reduce reinfection rates, beyond partner notification slips alone. Other options might involve healthcare professionals directly notifying partners or using electronic messaging to address disclosure concerns. The findings could also help determine which strategies are most suitable based on factors like partner type, the gender of the index patient, and the number of partners involved.”

23. Lines 244-245 mentioned a limitation of the current LUSTRUM framework when applied in South African culture. Please consider adding a separate ‘Implications’ paragraph where you reflect on this conclusion and state potential adjustments for future research. Thank you for this suggestion. We have included a implications paragraph that speaks to the discussion and limitations.

24. Lines 248-249: Could you indicate (statistical) evidence of representativeness of your study sample? We have elaborated on the statistical evidence of representativeness in lines 311-314; “The study prevalence reflected the estimating national population prevalence of 5% (3% among men; 7% in women). Therefore, a sample size of 900 participants was sufficient in determining the diagnostic performance and the target product profile requirements of the NG-LFA.”

25. A limitation in line 252 is the need for professional nurses when defining partner relationships. Could you explain if this is feasible in real-world applications of partner notification strategies? Thank you for raising this query. As part of the study, the partner classifications were part of a survey, that was administered by a senior field worker. For clinical management, a professional nurse that counsels on partner notification and dispenses antibiotics for STIs would be needed. We’ve clarified this in Lines 318-321:

“However, having professional nurses who counsel on STIs and partner notification, and provide treatment at the primary healthcare level, classify partner relationships at diagnosis (as in the LUSTRUM trial) could better inform future partner notification strategies (beyond partner classifications alone)”

REVIEWER # 2

26. The authors nicely outline their experience with assessing partner types as a part of a larger STI study. It would be helpful to better define the LUSTRUM partner definitions and explain how the partner types differed from partner type categorization to better orient the reader. It would also increase the impact of your work if you could further tie in your thoughts regarding how well the LUSTRUM definitions worked in your contexts and what steps if any would help propel the field forward in your context. Thank you for your interest in our paper, and we have taken note of your suggestions. We have made various revisions, to ensure partner definitions, types, and categorizations are understand, including how well LUSTRUM’s classification works in our context.

27. OVERALL

I recommend maintaining one term - partner notification vs. disclosure - throughout the manuscript for consistency. We have taken note of this suggestion. However disclosure (status sharing) by an index patient may have a different definition than partner notification (in the instance where a partner is notified without a partner disclosing their status e.g., electronic messaging, a healthcare provider might notify a partner). This reveals why relying on index patient disclosure for partner notification may not be effective depending on partner type. However, we have ensured that each term is used correctly for consistency.

28. ABSTRACT

Please specify the term “rates” in the first paragraph. From the

---

## [Editor Report · Decision Letter 1]

8 Apr 2025

Spectrum of sexual partner types among adults screened for sexually transmitted infections in the Eastern Cape, South Africa

PONE-D-24-26246R1

Dear Dr. Mdingi,

We’re pleased to inform you that your manuscript has been judged scientifically suitable for publication and will be formally accepted for publication once it meets all outstanding technical requirements.

Kind regards,

Caroline Watts, PhD

Academic Editor

PLOS ONE
---

## [Editor Report · Acceptance letter]

PONE-D-24-26246R1

PLOS ONE

Dear Dr. Mdingi,

I'm pleased to inform you that your manuscript has been deemed suitable for publication in PLOS ONE. Congratulations! Your manuscript is now being handed over to our production team.

Kind regards,

on behalf of

Dr. Caroline Watts

Academic Editor

PLOS ONE